# Dynamic Fragility of a Slender Rock Pillar in a Carbonate Rock Mass

Alaa Jbara[1], Michael Tsesarsky[1,2]

1 Department of Civil and Environmental Engineering, Ben Gurion University of the Negev, Beer-Sheva 84105, Israel
2 Department of Earth and Environmental Sciences, Ben Gurion University of the Negev, Beer-Sheva 84105, Israel

Correspondence to: Michael Tsesarsky (michatse@bgu.ac.il)

**Abstract.** Fragile geological features (FGFs) provide critical empirical data for validating probabilistic seismic hazard models over prehistoric timescales. Among FGFs, precariously balanced rocks (PBRs) are the most widely studied, with fragility analyses based on simple, rigid body, rocking dynamics. FGFs formed from sedimentary rock masses differ from PBRs and require consideration of rock mass properties in their fragility assessments. Sedimentary FGFs received limited attention from

the geological and engineering communities. This study presents a detailed dynamic fragility analysis of a 42-meter-high Ramon Pillar (Negev Desert, Israel). Composed of a sedimentary rock mass with various discontinuities, the pillar was modeled using a high-resolution finite elements (FE) model with $1.25 \cdot 10^6$ elements. The model was constructed using high-resolution aerial LiDAR scanning and in-situ measurements of rock elastic modulus along the pillars' height. Validation was achieved by comparing computational modal analysis with in-situ measurements of natural vibrations, accurately predicting

the first mode (1.3 Hz) and estimating the second mode (2.7 Hz) with a 10% deviation from observed values (3 Hz). The assumption of uniform rock elastic moduli (back-calculated) or simplified geometries yielded unsatisfactory results, highlighting the importance of precise characterization. Situated near two significant seismic sources, the Sinai Negev Shear Zone (SNSZ) with a potential M 6 earthquake and the Dead Sea Transform (DST) with a potential M 7 earthquake, both with sub-millennial return periods, the pillar's fragility was used to test regional seismic hazard estimates. Two methodologies were

employed: a simplified spectral analysis based on empiric ground motion models and a fully dynamic FE analysis incorporating recorded ground motions from the PEER strong motion database. Results show that an M7 on the DST (45 km away) will not comprise the pillar integrity, whereas an M6 earthquake on the SNSZ (6 to 20 km away) would likely lead to breakage at its base due to tensile stresses exceeding its basal strength. Given the pillar fragility age of 11.4 ky, these findings challenge the assumption that the SNSZ can produce an M 6 event.

## 1 Introduction

The recurrence intervals of large earthquakes generally exceed the observation length of instrumental records; hence, existing catalogs cannot provide complete information on seismic sources and seismic hazards (Anderson et al., 2011). Whereas historical records are useful in constraining the return periods, they are less useful in constraining earthquake locations, magnitudes, and ground motion intensity. The standard method used to assess the hazard of potentially damaging earthquakes

is the probabilistic seismic hazard analysis (PSHA). This framework allows for estimating the rate or probability of exceeding

ground-motion intensity at a site (Gerstenberger et al., 2020). Whereas the underlying assumptions of PSHA are still debated, (e.g., Mulargia et al., 2017; Stark, 2022; Bommer, 2022) its widespread use in the earthquake engineering community requires independent validation (Marzocchi and Meletti, 2024).

Fragile geological features (FGFs) provide critical empirical data for validating probabilistic seismic hazard models over prehistoric timescales. The development of robust and quantitative validation and evaluation methods to reduce uncertainties in earthquake ground-motion estimates are particularly required at long return periods ($10^3$ to $10^4$ yr) because PSHA estimates for such return periods are highly uncertain yet essential for the siting, design, and continued maintenance and monitoring of critical civic facilities, such as large dams, power plants (including nuclear), and nuclear waste repositories (Rood et al., 2020). Ancient, fragile geologic features (FGF) have been previously identified as potentially useful for validating un-exceeded ground motions estimated from PSHA models (Anderson et al., 2011 and references therein; Stirling and Anooshehpoor, 2006), and were recently incorporated into formal design earthquake motions for a significant engineered structure (Stirling et al., 2021).

## 1.1 Fragile Geological Features

A fragile geological feature (FGF) is a feature that might be easily destroyed by strong earthquake ground motions and is mechanically simple enough to analyze the ground motions that might cause its destruction (Anderson et al., 2011). FGFs include various delicate natural features such as paleo-sea stacks, tufa towers, hoodoos, badlands, and unstable regoliths, which can potentially be used to constrain past ground motions (Stirling et al., 2020). In practice, the most studied and used FGF is the precariously balanced rocks (PBR), introduced by Brune (1996) for PSHA applications in Southern California and Nevada. PBRs are boulders balanced on and mechanically separated from a sub-horizontal pedestal and are susceptible to topple when exposed to earthquake ground shaking (Hall et al., 2019; Rood et al., 2022; Chen et al., 2024; Mcphillips and Pratt, 2024). The ubiquity of granitic PBRs near the San Andreas fault system enabled statistically meaningful analysis and reduced uncertainties in earthquake hazard analysis (Rood et al., 2020). The relative simplicity in determining the fragility of PBRs enabled the spread of the method worldwide. However, this method cannot be readily exported to different geological terrains and lithologies as the mechanical response to the dynamic loading of various FGFs fundamentally differs from the rocking dynamics of PBRs. In a recent workshop on "*Evaluation of seismic hazard models with fragile geological features*" (Stirling et al., 2021) ,the topic of "*fragility estimation*" was recognized as "*a critical research*" need. Specifically, with emphasis on "*Case studies of the fragility of various categories of FGF*s" and "*cost- and time-effective methods for quantifying fragility that accounts for frequency content of ground motions*", that will yield "*greater confidence in fragility assessment*" and "*greater uptake of FGF data for constraining seismic hazard models*".

## 1.2 Dynamic analysis of FGFs

Contrary to rigid PBRs, FGFs with fragility depending on rock mass structure and properties received lesser attention from the geological and engineering communities. A common issue for non-PBR FGFs is the strength of rock mass discontinuities,

specifically across its base (basal attachment). For FGFs evolving from sedimentary rock masses, complete basal detachment along bedding planes cannot be assured. Shang et al. (2018) showed that in siltstone, an incipient bedding plane's uniaxial tensile strength (UTS) ranged from 32% to 88% of the parent rock UTS. Rock joints in the same rock type exhibited UTS of 23% to 70% of parent rock, and rock bridges consisted of 23% to 70% of the discontinuity. Frayssines and Hantz (2009) showed that steep limestone cliffs were stable due to relatively small rock bridges, up to 5% of the failure surface. Any attempt to quantify rock bridges is exacerbated by the fact that rock bridges are not visible unless human activities or natural events expose the rock mass (Elmo et al., 2018). Should rock bridges be neglected in hazard assessment, the analysis would conclude that cliffs that existed from the centennial to millennial time scales have low safety factors. Therefore, a realistic fragility assessment of rock pillars should consider the basal attacment and its tensile strength.

Accurate estimation of natural frequencies, elastic moduli, and damping ratios is critical for assessing the dynamic fragility of freestanding rock structures. Several research groups studied the vibrational behavior of freestanding rock masses, such as rock arches (Moore et al., 2018; Moore et al., 2016) and rock towers (Moore et al., 2019; Valentin et al., 2017). Combined with numerical analysis to back-calculate the elastic (small strain) moduli, the seismic resonance technique proved feasible to determine the natural modes and the elastic moduli of freestanding rock masses. Specifically, Moore et al. (2019) claim that with basic geometry and material properties estimates, other freestanding rock structures' resonant frequencies can be estimated a priori. It should be noted that installing seismometers atop large-scale structures, such as the 120 m high Castleton tower in Utah (Moore et al., 2019), is not a simple task involving rock climbing and rappelling expertise.

In this study, we present a comprehensive analysis of the dynamic fragility of a slender rock pillar (Ramon, Israel) based on accurate LiDAR scanning of its geometry, in-situ rock elastic modulus determination, and finite element (FE) modal and dynamic analysis. The mechanical model was validated by comparing the results of the modal analysis to vibrational measurements of the pillar (Finzi et al., 2020), ensuring its reliability. Leveraging the validated model, we performed a fully dynamic FE analysis under various seismic loading (distance – magnitude) scenarios. The dynamic analysis results are used to challenge previous assumptions regarding the region's seismic hazard. This research shows the significance of accurate and detailed material models of sedimentary structures for a dynamic fragility analysis and provides a straightforward approach for their analysis as FGF. The findings highlight the potential of this approach in advancing seismic hazard assessment methodologies and improving the understanding of the fragility of sedimentary rock formations.

## 2 Negev Desert Seismic Hazard

The main seismic source in the eastern Middeteranena is the active tectonic border of the Dead Sea Transform (DST). The DST, with a total length of 1100 km, consists of several en-echelon segments extending from the Gulf of Aqaba in the south to the Syrian-Turkish border in the north (Garfunkel, 2014). The seismic hazard of the central Negev (Israel) is dominated by two seismogenic sources: the Sinai-Negev Shear Zone (SNSZ) and the southern part of the DST. The SNSZ includes five E–W trending faults (Fig. 1), from north to south: Saad–Nafha, Ramon, Arif–Batur, Paran, and Thamad. The region exhibits low

modern-day seismicity. Seismic hazard studies regard the entire zone cautiously by setting a relatively high maximal magnitude of M 6.2 (Grünthal et al., 2009). They also report a common b-value of 0.838 ± 0.022 for the DST to be used for small seismic source zones within the large zone. The annual rate for an M 6.2 earthquake is in the order of $10^{-2}$. Shamir et al. (2001) and the Israeli earthquake building code consider only the Thamad and Paran faults as seismic sources capable of producing M 6 earthquakes. The Arava fault, a relatively simple linear segment of the DST, is characterized by almost pure strike-slip motion with a slip rate of about 4-5 mm/year (Hamiel et al., 2016). Grünthal et al. (2009) report a b-value of 0.904 ± 0.60 for the Dead Sea area faults. While the DST (at large) has been responsible for numerous M ≥ 7 earthquakes in the last 3000 years (Agnon, 2014; Zohar, 2019), only a few significant events were documented along the Arava fault section (Lefevre et al., 2018). There is a wide range of uncertainty regarding the magnitudes of earthquakes along this remote section. The challenge in deciphering historical data is demonstrated in the case of the 873 CE event, which some studies suggest to be the strongest historical earthquake in the region, M 7 to 7.5 (Lefevre et al., 2018; Klinger et al., 2015), while other studies do not mention it at all (Ambraseys et al., 2005).

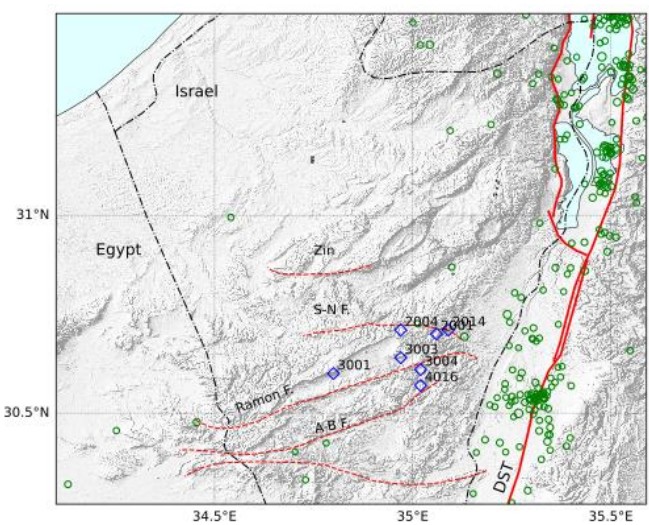

**Figure 1. Regional shaded relief map of Southern Israel. Dead Sea Transform (DST) active faults are plotted with continuous red lines, and the Sinai Negev Shear Zone faults are plotted with broken blue lines. S-N F. is Sa'ad-Nafha Fault, A-B F. is Arif-Bator fault. Blue diamonds are the locations of dated rock pillars (Finzi et al., 2020). Green circles are seismic events (Israel Seismic Bulletin, 2013 – 2023, M > 2).**

### 2.1 Negev Desert FGFs

Finzi et al. (2020) conducted an extensive survey of fragile geological features (FGF) in the Negev Desert, documenting over 80 FGFs, half of which are rock pillars. For nine rock pillars, the fragility age was determined using the Optical Stimulated Luminescence (OSL) technique, ranging from 123 ky to 1.7 ky. The Negev rock pillars form along pre-existing fracture sets that cut vertically into cliff-forming layers of the hard carbonates of the Judea group. As the fractures grow and widen, they separate rock columns from the cliff (Frayssines and Hantz, 2009; Bakun-Mazor et al., 2013), eventually evolving into

freestanding pillars (Fig. 2). The erosional slope retreatment rate in this hyperarid ( < 80 mm/y precipitation) area is slow, about 10m Ma$^{-1}$ (Boroda et al., 2014), prevailing since the middle Pleistocene (Enzel et al., 2008). The long-term climatic stability and proximity to seismic sources make the Negev rock pillars excellent candidates for studying their dynamic fragility and testing basic assumptions of regional PSHA.

Among the rock pillars mapped in the Negev Desert, the Ramon Pillar (#3001 in Finzi et al., 2020), located at the northern cliff of Ramon crater (30.606N 34.804E), is the most impressive (Fig. 2). The pillar's height is 42 m, with a slenderness ratio (height/width) of 8, comprised mainly of hard carbonates of the upper Cretaceous Hevyon Fm. The fragility age of the pillar, based on OSL dating of silt accumulated in the large crack separating it from the cliff, is 11.4 ky (Finzi et al., 2020).

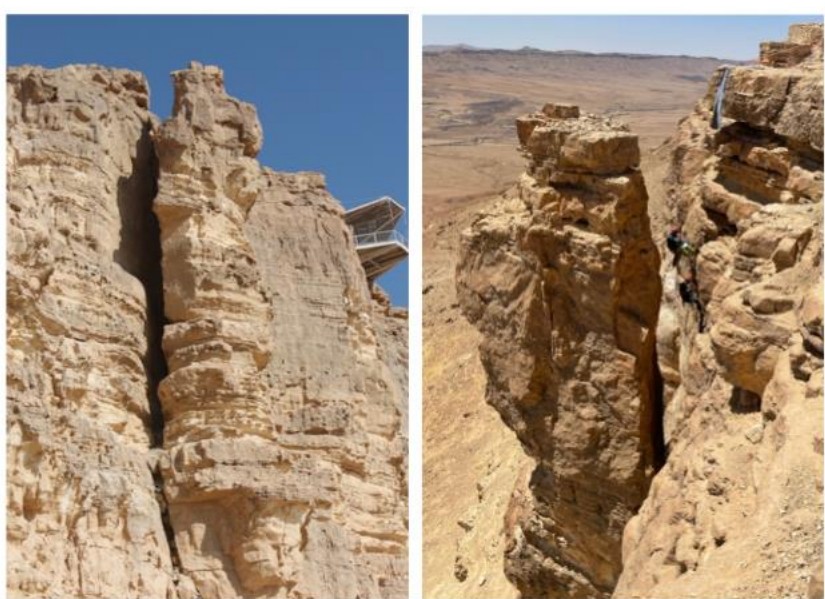

**Figure 2. The Ramon pillar: a) photo taken from the Ramon crater floor, looking northeast (left); b) rappelling along the back crack for measurement of rock elastic modulus, looking southwest (right).**

## 3 Methods

### 3.1 Pillar Geometry and Rock Mass Properties

The Ramon pillar was scanned using airborne LiDAR (Geoslam ZebHorizon sensor) with a 2 - 4 cm accuracy. A solid model was rendered from the point cloud using the AutoDesk MeshMixer V. 3.5 (https://meshmixer.com/) software bundle. In this study, the various discontinuities within the rock mass were not explicitly modeled. Instead, their influence was incorporated indirectly by adjusting key rock mass properties, specifically stiffness and damping, to account for their presence.The density of the hard carbonate is 2,230 kg/m$^3$, and tensile strength (Brazilian splitting test) ranges from 5 to 9 MPa (Saltzman, 2001). The rock's elastic modulus was estimated from direct measurements of elastic rebound at the back of the Ramon pillar (Fig.

2b). The team led by co-author Tsesarsky rappelled the entire length of the pillar back joint (42 m), taking measurements using a rebound hammer (Proceq Rock Schmidt) at discrete locations along the pillar's height to capture lithological and mechanical

variations. The Schmidt hammer rebound hardness was determined following the ASTM C805/C805M-18 standard. Measurements focused on transitions between lithological units, such as from hard dolomites/limestones to softer limestones (e.g., fossil reefs) and within limestones transitioning from massive to bedded structures. For instance, at a depth of 12 m, the rock consists of massive limestone, transitioning to a porous fossilized reef at 15 m (refer to Fig. A1) and subsequently to uniform limestone. Due to lithological heterogeneity and the physical constraints of rappelling, the resulting hardness profile

is not evenly spaced. Details of the sampling layout are provided in Table A1 of the supplementary material. The Katz et al. (2000) correlation was applied to estimate the rock elasticity modulus. Rock mass modulus ($E_{RM}$) was assessed using the Hoek and Diederichs (2006) equation and a GSI value of 65 based on a close visual inspection of the rock mass. The variation of rebound value and $E_{RM}$, along the height of the pillar, is presented in Fig. 3. Based on these measurements, visual inspection, and geological judgment, the pillar was divided into four vertical regions, each with a representative rock mass modulus,

presented in Fig. 3 as vertical gray lines.

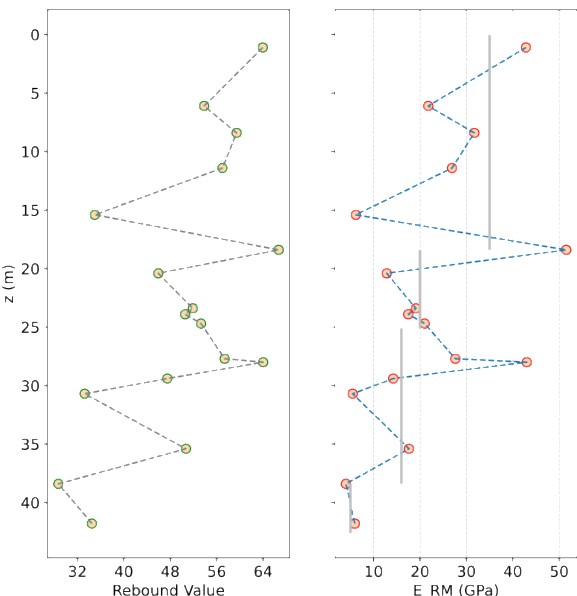

**Figure 3. Variation of rock rebound value (left) and rock mass elastic modulus (right) along the Ramon pillar back crack, z is the depth from the surface. Gray lines are representative values for the four vertical regions used in the FE model.**

### 3.2 FEM Analysis

Finite element analysis was performed using the ABAQUS software package (Simulia, 2020). The solid MeshMixer model was imported into ABAQUS and, due to the structure's complex geometry, meshed using tetrahedral linear elements. The pillar was modeled as an isolated free-standing structure, excluding the parent cliff of the Ramon Crater from the computational domain. During the gravity loading phase, fixed boundary conditions were applied at the base, constraining the translational

and rotational degrees of freedom. The boundary conditions for the dynamic loading phase were adjusted to zero vertical
displacements at the base while applying time-dependent horizontal accelerations to simulate seismic excitation. The model
was horizontally loaded in one direction at a time by one horizontal component. H1 was loaded in the 135° direction and H2
in the 45° direction.

Convergence analysis was performed to set the optimal element size, resulting in 0.5 m elements. In total, the model comprised
$1.248 \cdot 10^6$ elements. For modal analysis, material properties were changed between models from simple homogenous models
to models with vertical changes in $E_{RM}$. For all the models, rock density was set to 2,230 kg/m$^3$, and the Poisson ratio was
assumed as 0.25. Dynamic models were executed using implicit formulation. Loading time histories were selected from the
Pacific Earthquake Engineering Research Center (PEER) ground motion database (Ancheta et al., 2013) for selected
magnitudes and distances.

## 4 Results

### 4.1 Modal Analysis

Modal analysis of the Ramon pillar with different levels of complexity was performed, starting with an equivalent homogenous
cylinder and followed by higher complexity models (refer to Table 1). The "Simplified" model was based on simplified
geometry developed from selected cross-sections extracted from photogrammetric scans and interpolated using AutoCAD
software. The "Scan_H" and "Scan_M" models were based on high-resolution LiDAR scans. These two models differ in
assigned $E_{RM}$, assumed homogenous for the H model and measured for the M model. Our analysis focuses on the first two
modes of the pillar (refer to Table 1), as they collectively account for over 80% of the modal mass participation (Chopra,
2014). Visualization of the modal analysis for the Scan_M model is presented in Fig. A2.

**Table 1. Results of the Ramon Pillar Modal Analysis**

| Model | $E_{RM}$ (GPa) | Mode 1 (Hz) | Mode 2 (Hz) |
|---|---|---|---|
| Equivalent Cylinder | 10 | 1.4 | 7.9 |
| | 20 | 2 | 11.2 |
| Simplified | 10 | 1.76 | 2.49 |
| | 20 | 2.22 | 3.14 |
| Scan_H | 10 | 1.5 | 2.1 |
| | 20 | 2.9 | 4.22 |
| Scan_M | In-situ | 1.3 | 2.71 |
| Measured | | 1.32 | 3.1 |

These results were compared with measurements of the natural vibration modes reported by Finzi et al. (2020). The original measurements were planned and supervised by co-author Tsesarsky. A broadband seismometer (Geospace GS-1) was positioned on the top of the pillar to record the ambient vibrations. The data were processed using a typical seismic noise data analysis workflow: The instrumental response was removed, and data were detrended and band-pass filtered between 0.1 and 30 Hz (the original recording was performed at 100 Hz). The results of these measurements are presented in Fig. 4 for completeness. The revisited data also contains corrected orientation data.

The first two modes of the pillar are bending modes at 1.32 Hz and 3.1 Hz, clearly visible on the horizontal components. The first mode is bending over the thinner horizontal dimension, normal to the cliff direction (135°), whereas the second mode is bending parallel to the cliff (45°) over the thicker horizontal dimension. The horizontal-to-vertical ratio of the vibrations can be used to define the prominence of the modes, 18 and 13 for the first and second modes, respectively. Higher modes, at 7.5 Hz and 11.1 Hz, are also clearly visible in the two horizontal components. Damping ratios were determined using the "half power bandwidth" method (Chopra, 2014). Using the horizontal to vertical channel ratio, the damping ratios are 6% and 4% for the 135 and 45 components, respectively.

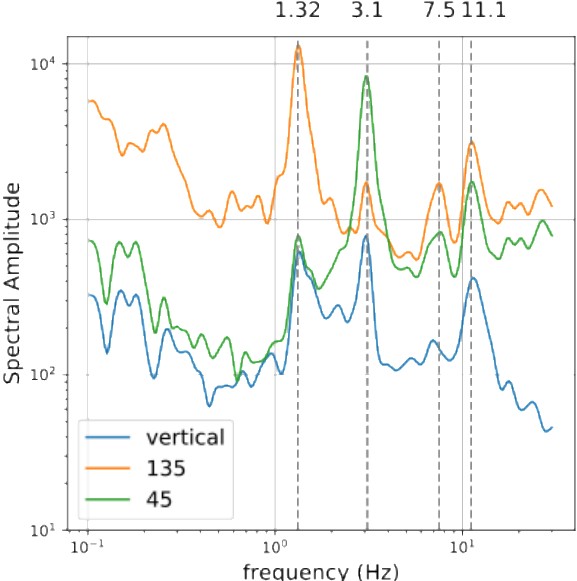

**Figure 4. Spectral amplitudes of the Ramon Pillar natural vibrations.**

As expected, the calculated modes are determined by the interplay between the rock mass elastic modulus ($E_{RM}$) value, here changed from 10 GPa to 20 GPa, and the model geometry. Using an equivalent cylinder model shows considerable discrepancy between the measured and calculated modes, specifically in the second mode. Moving forward to a simplified yet more representative geometrical model improves the prediction; however, it still has low accuracy. The two scanned models exemplify the difference between assumed and measured $E_{RM}$. For the "Scan_H" model, assuming $E_{RM} = 10$ GPa yields a good

approximation of the first vibration mode: 1.5 Hz compared with measured 1.32 Hz. However, the second vibration mode is considerably lower than measured, 2.1 Hz compared with 3.1 Hz. Assuming $E_{RM}$ = 20 GPa increases the discrepancy in both modes. Using the same geometry, however, with in-situ determined $E_{RM}$, including vertical variations, yields the most accurate results. The first vibration mode is 1.3 Hz, measured and calculated, and the second mode is 3.1 Hz measured compared to 2.7

205  Hz calculated. The discrepancy between the measured and modeled in the second mode suggests that the model could be further refined, specifically the vertical distribution elastic modulus. The accurate predictions of the natural vibration modes are considered to be the validation of our FE model for the Ramon pillar and use this model for dynamic analysis.

### 4.2 Simplified Fragility Analysis

Prior to conducting the computationally intensive finite element method (FEM) dynamic analysis, the fragility of the Ramon

Pillar was assessed using a simplified approach. For this analysis, a maximum magnitude earthquake was defined for each causative fault at the closest distance to the Ramon Pillar (refer to Table 2). The pseudo-spectral accelerations (PSA) for each scenario were calculated using the ASK14 ground motion model (Abrahamson et al., 2014). ASK 14 was selected as the representative of NGA-2 West-based GMMs. Comparison of the different NGA-2 West-based GMMs (Gregor et al., 2014) for the studied magnitude range shows little variation between the different models. This research focuses on the median PSA

and one standard deviation (1σ) to analyze the stresses. In Deterministic Seismic Hazard analysis, it is typically assumed that one standard deviation (84th percentile) represents the maximum (worst-case scenario) ground motion, which is assumed to be the boundary between physically possible and unphysical ground motions (Strasser et al., 2008). Whereas such a limit is not imposed in PSHA the truncation depends on return periods. Assuming an annual frequency of exceedance (EFA) of $10^{-3}$ (millennial return), the difference between 1σ and 2σ – 3σ is relatively small, large deviations are expected for EFA of < $10^{-4}$

. Figure A3 in Appendices presents the PSA of the different scenarios. Assuming that the pillar behaves as a cantilever, which was proven by the in-situ measurements and modeling, the moment and the tensile stresses at the base of the cantilever were calculated using an equivalent cylinder with R = 3.8 m. For the calculation, an elastic modulus of 13 GPa was assumed, which gave the best compliance with the first natural mode of 1.32 Hz. The results of the simplified analysis are presented in Table 2.

For each of the scenarios studied, the median tensile stress at the base of the pillar is below the tensile strength of the rock. The highest value is 2.5 MPa for the Ramon fault scenario, and the lowest value is for the Paran fault scenario, 1 MPa. These values are lower than the tensile strength of the rock (5 – 9 MPa). Assuming a 50% strength reduction due to incipient discontinuities will bring the tensile stresses at the base to the lower bound of rock strength only for the Ramon fault scenario. Taking into account one standard deviation (1SD) the stresses at the base of the pillar are typically doubled. For the Ramon

scenario, the tensile stress is 4.6 MPa, and for the Nafha-Saad and Arava scenarios, it is 3.8 MPa, bringing the stresses at the base of the pillar close to failure.

**Table 2. Results of the Simplified Fragility Analysis for an equivalent cylinder (R = 3.8m). PGA is peak ground acceleration; values in parentheses are plus one standard deviation. SA @f₁ is the median spectral acceleration at the first natural mode, SD is the standard deviation, and σt is the median tensile stress at the base of the pillar. PGA and SA were calculated using the ASK14 ground motion model.**

| Scenario | Causative fault | PGA (g) | SA @$f_1$ (g) | SA @$f_1$ + SD (g) | $\sigma_t$ (MPa) | $\sigma_t$ +SD (MPa) |
|---|---|---|---|---|---|---|
| M 6 R 6 | Ramon | 0.21 (0.40) | 0.13 | 0.26 | 2.5 | 4.6 |
| M 6.2 R 10 | Nafha-Saad | 0.15 (0.30) | 0.09 | 0.18 | 1.9 | 3.8 |
| M 6.2 R 26 | Paran | 0.06 (0.12) | 0.04 | 0.08 | 1 | 1.8 |
| M 7.5 R 45 | Arava | 0.08 (0.15) | 0.08 | 0.17 | 1.8 | 3.6 |

## 4.3 Dynamic Analysis

The dynamic analysis focuses on loading scenarios typical to the seismic sources of the region: M 6.2 on the SNSZ faults and M 7 on the DST. For the SNSZ, the analysis focused on $R_{RUP}$ < 10 km, representing earthquakes on the Ramon and Saad-Nafha faults. For the DST, $R_{RUP}$ of 45 km was assumed, representing the shortest distance to the Arava fault. Time series for the dynamic analysis were obtained from the PEER strong motion database, using faulting style, magnitude, distance, and site criteria. These criteria are assumed to serve as a fundamental analogy for the faults under consideration. Names of the events and ground motion parameters used in our analysis are presented in Table 3. For each event, the two horizontally perpendicular loading components were simulated.

The first step of our analysis was to study the effect of damping on the pillar's stresses and displacements. Damping ratios for freestanding rock structures depend on geometry, mass and stiffness distribution, degree of continuity, and other attributes of natural rock masses. Finnegan et al. (2022), and references therein show that for sandstone freestanding rock structures, the damping ratio ranges from 1% to 3%; however, higher values of 8% to 10% have been reported for heavily jointed rock masses. Our estimate for the Ramon pillar falls within the reported range, with a damping ratio of about 5%, reflecting the discontinuous nature of the rock mass. In Abaqus, a Rayleigh damping of 2%, 5%, and 7% was used to study the variations of the tensile stress at the pillar's base and the displacement at the top. For this analysis, the Morgan Hill M 6.2 earthquake was utilized (refer to Table 2). As expected, the maximal tensile stress and the displacements diminished with damping. The tensile stresses reduced from 15.6 MPa for 2% to 10.3 MPa for 5% and 8.65 MPa for 7% damping. Respectively, the displacement at the top of the pillar reduced from 0.07 m to 0.04 m and 0.037 m. The sensitivity to the damping ratio is maximal when changing the

value from 2% to 5%, the tensile stress value changes by 34%, and the displacement by 57%. Further increasing the damping ratio results in considerably lesser changes. A damping value of 5% was selected for the dynamic analysis, based on the measurements and the results of the sensitivity study.

A typical result of the dynamic analysis, Morgan Hill 1984 M 6.2 earthquake (MH) is presented in Fig. 5. Results for selected elements at the base are presented in the appendix, Fig. A4. Please recall that each component is loaded in one direction at a time, representing the thinner and thicker dimensions of the pillar, respectively. H1 is loaded in the 135° direction and H2 in the 45° direction. Dynamic loading induces a transition in elements from compressive to tensile stress (and strain) regimes, with boundary elements experiencing the most significant changes. For the MH earthquake, the dynamic loading is almost symmetric (refer to Fig. 5), resulting in almost symmetric changes in dynamic stresses. However, due to the pillar's geometry, the initial (gravity) stress distribution at the base is nonuniform. Elements initially subjected to compressive stresses (e.g., element 115637 in Fig. A4) experience increased compressive stresses during dynamic loading, while those initially in tension experience increased tensile stresses. In our analysis, maximum compressive stresses exceeded maximum tensile stresses by up to 25%. However, the compressive strength of most rocks typically surpasses tensile strength by a factor of three(Yu et al., 2020) to ten (Sheorey et al., 1989). Consequently, the limiting strength condition is tensile, which is the primary focus of this study.

For the MH earthquake and all other earthquakes modeled, each loading component results in different stresses at the base and displacements at the top of the pillar, reflecting the loading time history (amplitude and duration) and the direction of loading. The main difference between the two horizontal components is the amplitude of the surface waves. In the H1 direction, the amplitude of the surface waves is considerably lower than the amplitude of the shear waves, whereas, in the H2 direction, the surface wave amplitude is in the same order as the amplitude of the shear waves (please refer to Fig. 5). The strong "jolt" of the surface waves in the H2 direction results in a 30% increase of the tensile stresses at the pillar's base, from 10.4 MPa to 13.3 MPa.

The results of the dynamic analysis are summarized in Table 4. The highest tensile stress value at the pillar's base is 17.68 MPa for the Chi-Chi M 6.2 H1 component. The lowest tensile stress value at the pillar's base is 5.85 MPa for the Parkfield M 6 ($R_{RUP}$ = 10 km) H1 component. It should be noted that these two extremes reflect the different faulting styles between the

two events; whereas the Chi-Chi event (an aftershock of the $M_w$ 7.7 event) is reverse faulting, the Parkfield event is a strike-slip. Typically, reverse faults produce stronger ground motions than strike-slip faults (such as SNSZ or DST). However, we wanted to study the dynamic behavior of the pillar under various loading scenarios with different PGA, IA, and duration. The scenario of a remote M 7 earthquake, represented by the 1999 Duzce M 7.2 event, results in relatively low tensile stress and

displacement: 3.92 MPa and 0.003 m, respectively.

**Table 3. Earthquakes catalog used in the dynamic analysis of the Ramon Pillar. M is the magnitude, $R_{RUP}$ is the distance to rapture, PA is the peak acceleration, IA is the Arias intensity, and $t_{595}$ is the significant duration of the event. H1 and H2 are the horizontal components. Ground motion time histories were downloaded from the PEER strong motion database (Ancheta et al., 2013).**

| Event | Mw | $R_{rup}$ (km) | PA H1 (g) | IA H1 (m/sec) | $t_{595}$ H1 (sec) | PA H2 (g) | IA H2 (m/sec) | $t_{595}$ H2 (sec) |
|---|---|---|---|---|---|---|---|---|
| Morgan Hill 1984 | 6.2 | 10 | 0.22 | 0.4 | 7.30 | 0.29 | 0.8 | 6.5 |
| Parkfield 2004 (a) | 6 | 10 | 0.15 | 0.15 | 9.95 | 0.17 | 0.17 | 11.04 |
| Parkfield 2004 (b) | 6 | 6 | 0.79 | 1.2 | 3.19 | 0.43 | 0.42 | 6.9 |
| Chi-Chi 1999 | 6.3 | 6 | 0.34 | 1.5 | 5.81 | 0.32 | 1.3 | 7.33 |
| Duzce 1999 | 7.2 | 45 | 0.03 | 0.008 | 23.20 | 0.02 | 0.005 | 26.3 |

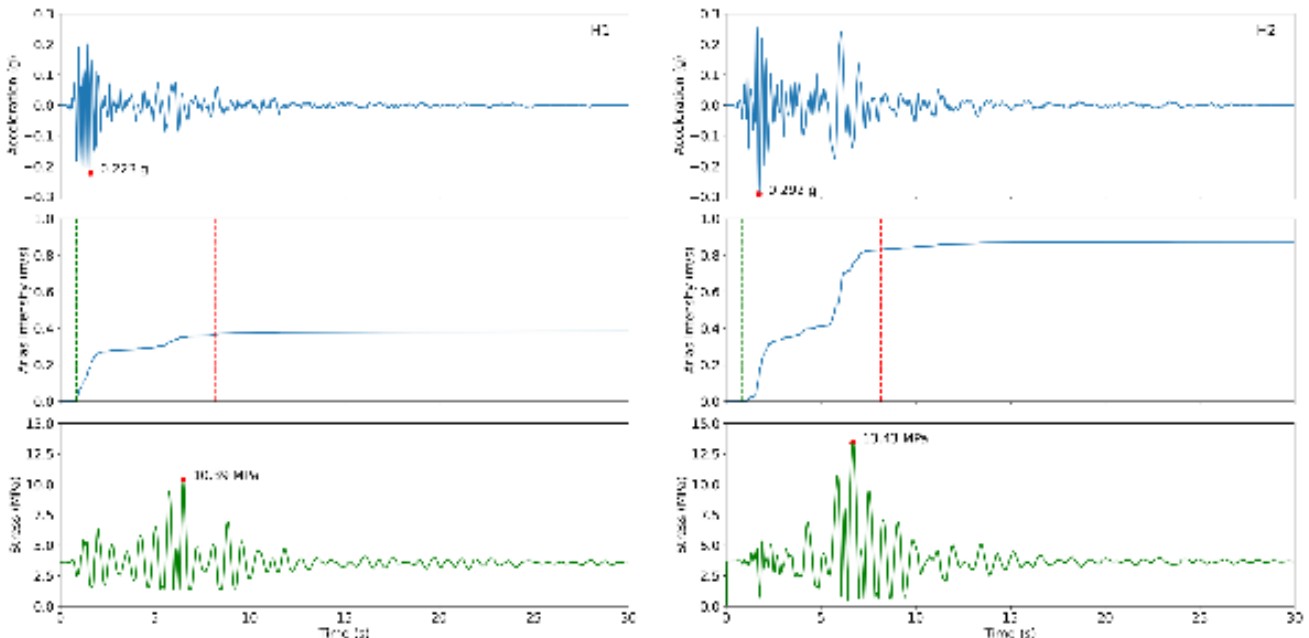

**Figure 5. Results of the Ramon Pillar dynamic analysis for the Morgan Hill 1984 earthquake with M 6 and $R_{RUP}$ = 10 km. The left panel is the H1 loading component, and the right panel is the H2 loading component. From the top: acceleration time history, Arias Intensity (vertical lines are t5 and t95 values), and maximal tensile stress at the bottom of the pillar.**

**Table 4. Horizontal displacement (δh) at the top of the pillar and tensile stress (σt) at the bottom of the pillar for the ground motions modeled. H1 and H2 are the horizontal components of ground motion.**

| Event | M | $R_{rup}$ (km) | $\sigma_t$ H1 (MPa) | $\delta_T$ H1 (m) | $\sigma_t$ H2 (MPa) | $\delta_T$ H2 (m) |
|---|---|---|---|---|---|---|
| Morgan Hill 1984 | 6.2 | 10 | 10.39 | 0.044 | 13.43 | 0.039 |
| Parkfield 2004 (a) | 6 | 10 | 6.02 | 0.023 | 5.85 | 0.012 |
| Parkfield 2004 (b) | 6 | 6 | 7.88 | 0.035 | 5.87 | 0.012 |
| Chi-Chi 1999 | 6.3 | 6 | 17.86 | 0.085 | 14.30 | 0.04 |
| Duzce 1999 | 7.2 | 45 | 4.18 | 0.01 | 3.92 | 0.003 |

## 5 Discussion

PBR stability analysis implicitly assumes that a hard, discontinuous contact (no moments resistance) exists between the base and the pedestal. In sedimentary rock masses, such as the Cretaceous carbonates of the Northern Negev, this assumption is not satisfied as many discontinuities, and beddings specifically, contain rock bridges with considerable tensile strength. Our measurements of natural vibrations of the Ramon pillar and subsequent FE modal analysis show that the pillar behaves as a cantilever. Static FEM analysis pillar shows that due to eccentric geometry and irregular geometry, the maximum tensile stress at the base of the pillar is 3.6 MPa. This value can be regarded as the lower bound of basal strength, including incipient discontinuities and rock bridges. Compared with the rock tensile strength, 5 to 9 MPa, this value represents a 56% to 20% strength reduction of the laboratory strength.

A simplified analysis based on an equivalent cylinder model and loads based on the ASK14 GMM shows that for the scenarios studied, only the Ramon fault (M 6 and $R_{Rup}$ = 6 km) can induce tensile stresses high enough to overcome the basal strength (refer to Fig. 6). For the median load, the basal tensile stress, 2.6 MPa, is well below the strength, and only for the single standard deviation load, the tensile stress is 4.6 MPa, which is higher by 20% only than the basal strength. Other scenarios yielded stresses lower than basal strength. Extending the analysis to $2\sigma$ truncation ground motion values elevate the SA and stresses at the pillar's base. For the Ramon scenario, basal stresses rise to 9.4 MPa, and for the Nafha-Saad scenario (M6.2 R10), the basal stress is 7.5 MPa. For the other two scenarios, the stresses are below the static value. The analysis presented here uses the ergodic ground motion model; a non-ergodic model (accounting for path and site effects) should be used if available. However, non-ergodic models for this region are not available yet.

Assuming that the basal strength was exceeded for the Ramon fault scenario and that the pillar is entirely disconnected requires a renewed analysis of the pillar in terms of PBR. Finzi et al. (2020) report a critical acceleration of 0.12g and dynamic

acceleration (Anooshehpoor et al., 2004) of 0.16g for the toppling of the Ramon pillar, assuming entirely discontinuous basal

conditions. For the Ramon scenario, the median PGA of 0.21 g is larger than the dynamic value, and for the Nafha-Saad scenario, it is slightly lower. For the Paran and Arava scenarios, median PGA values, 0.06 and 0.08 g, respectively, are considerably lower than the dynamic acceleration value. The 1σ PGAs are larger than the dynamic acceleration, rendering the Ramon and Nafha-Saad scenarios unstable. However, the Paran and Arava scenario's PGAs are still lower than the dynamic acceleration required to topple the pillar.

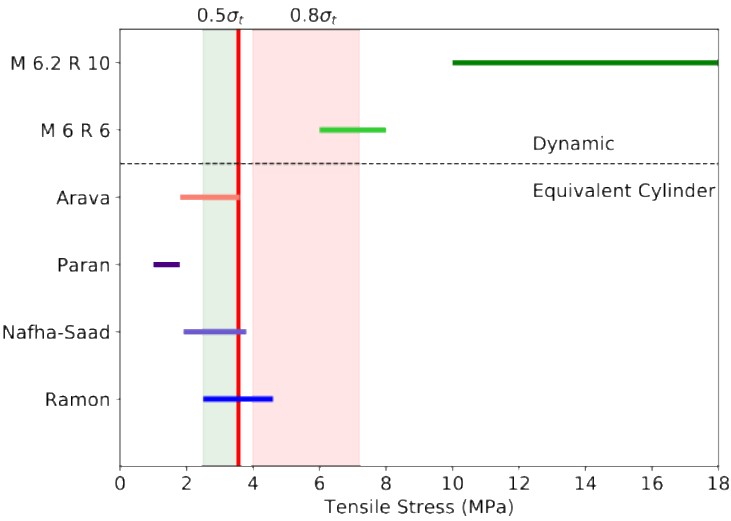

**Figure 6. Compilation of basal stresses for the Ramon Pillar analysis. For the Equivalent Cylinder models, the horizontal bar spans the tensile stresses for the spectral acceleration's median (low) and single standard deviation (high) at 1.32 Hz. For the Dynamic analysis, the horizontal bar spans the minimum and maximum tensile stresses. The vertical line at 3.6 MPa represents the maximum static basal stress. Shaded regions span the 80% (pink) and 50% (green) range of laboratory rock tensile strength.**

**5.1 Dynamic Fragility**

The equivalent cylinder analysis is based on a time-invariant, single-value (PSA) determination of basal stresses. However, since dynamic loading is inherently time-dependent, a fully dynamic analysis was conducted using the validated finite element (FE) model. The selected time series represent a range of acceleration and duration values (Table 3), from 0.15 to 0.75 g and 3 sec to 10 sec. Naturally, the modeled ground motions do not encompass a full suite of accelerations and durations but show

general trends.

Clearly, the dynamic analysis results in higher basal stresses. The minimal increase in tensile stresses was calculated for the Parkfield 2004 event (PGA of ~0.2g and IA of ~0.2 m/sec with duration of ~ 10 sec) to a value of 6 MPa, a factor of two higher than the static basal strength. The Morgan Hill 1984 event (PGA of ~0.3g and IA of ~0.4 to 0.8 m/sec with a duration of ~ 7 sec) increased the tensile stresses to 10 MPa and 13 MPa. The highest stresses were calculated for the M 6.2 Chi-Chi 1999 (an

aftershock of the $M_w$ 7.7 event) with 14 MPa and 18 MPa values in the two horizontal components.

Interestingly, the pulse-like loading of the Parkfield (b) event at $R_{Rup}$ = 6 km, with a PGA of 0.7 g and a duration of 3 sec, yielded similar displacement and stress values to the Parkfield (a) event (PGA = 0.17 g and duration of 11 sec) showing the detrimental effect of lower PGA with higher duration. Loading the pillar by a larger yet distant earthquake, represented by the M 7.2 Duzce 1999 earthquake, at $R_{RUP}$ = 45 km (not shown in Fig. 6), bares little effect on the pillar, elevating the stresses by only 17% and 9% for the two horizontal components above the static values. It should be noted that the records used in our dynamic analysis are from a global database. Due to different source, path, and site conditions, local time series are expected to differ in ground motion intensities (acceleration, velocity, duration, etc.). However, many regions globally lack instrumental coverage, or the recurrence intervals are long; hence, using global records is a necessary step towards a better understanding of the dynamic fragility of FGFs.

In this research, the pillar was modeled as a continuous cantilever structure fixed at its bottom. The accurate forward calculation of the natural vibration modes performed in this research supports the validity of our model and modeling approach. The rock mass's discontinuous nature was incorporated in the GSI rating for calculating rock mass modulus. Under this assumption, the stresses at the pillar's base are maximal, as neither frictional sliding or rocking is allowed, and the only energy dissipation is through the viscous Rayleigh damping. A damping value of 5% was employed, consistent with the measured damping.

Modeling sliding and rocking in FE is challenging as incorporating discontinuities into the continuous model is not trivial. An alternative approach is utilizing Discrete Element Methods (DEM) to study dynamic fragility. However, numerous numerical controls, such as penalties and frictional properties, are not easily calibrated or measured in situ. Furthermore, assessing the amount and distribution of rock bridges across bedding and joints is a non-trivial task. It should be recalled that even 5% of rock bridges stabilize cliffs of carbonate rocks (Elmo et al., 2018).

**5.1 Implications for Seismic Hazard**

The Ramon pillar is not sensitive to loading from strong and remote earthquakes (M 7 and $R_{RUP}$ > 45 km) and, therefore, cannot be used to constrain the seismic hazard from the DST. However, it was found to be sensitive to moderate and close earthquakes (M 6 and $R_{RUP}$ < 10 km) originating on the SNSZ. The equivalent cylinder approach and GMM-based load results in a non-conservative estimate of basal stresses. Fully dynamic analysis yields considerably higher stresses and indicates that the Ramon Pillar is sensitive to close earthquakes.

All of the M 6 earthquakes modeled dynamically induce tensile stresses at the pillar's base that are higher than its basal strength. The first exceedance of the strength typically occurs in the first seconds of the loading, typically within 25% of the loading duration ($t_{595}$), well before reaching the peak stress. Thus, it can be assumed that a bedding plane with rock bridges will fail during loading, leading to detachment of the base. Under fully discontinuous conditions and assuming rocking mechanics (PBR type), the required dynamic acceleration to topple the pillar is 0.16g, well within the PGA range for the magnitude-distance of the SNSZ faults.

Based on the analysis, it is postulated that an M 6 event on the Ramon and Nafha-Sa'ad faults didn't occur during the pillar's fragility age over the past 11,000 years. Our analysis challenges the assumption that the SNSZ as a whole can produce an M

6.2 earthquake. To determine whether the southern Paran fault is capable of an M 6.2 earthquake, a closer FGF should be analyzed.

The analysis presented here is the first step to constrain the seismic hazard on the SNSZ, and more FGFs in the region should be analyzed for better temporal and spatial coverage. The work of Finzi et al. (2020) lists nine pillars in this region (including the Ramon pillar) with fragility ages ranging from 1.4 ky to 123 ky. It should be recalled here that typically, the number of FGFs used for constraining PSHA is low; please refer to recent examples of Rood et al. (2024) and Stirling et al. (2021).

## 6 Conclusions

This research studied the dynamic fragility of a 42 m high, slender rock column comprised of discontinuous sedimentary rock mass located on the rim of the Ramon erosional crater (Israel). The pillar is found near two seismic sources: The Sinai Negev shear zone (SNSZ) and the Dead Sea transform (DST).

The pillar was aerially scanned with high-precision LiDAR. Rock mass elastic stiffness was measured in situ by rappelling the entire length of the pillar. Based on the scan and measurements, a finite element (FE) model for the pillar was developed.

The FE model was validated by comparing the modal analysis (assuming cantilever boundary conditions) results to the in-situ measured vibrational modes of the pillar. The comparison shows that the first two modes are highly compatible: 1.3 Hz and 3 Hz measured vs. 1.3 Hz and 2.7 Hz modeled.

We first studied the pillar's fragility using a simplified approach based on pseudo-spectral accelerations and an equivalent cylinder with R = 3.8 m. For the different scenarios studied, only the M 6 and $R_{RUP}$ = 6 km scenario yielded basal stresses exceeding the pillar's basal strength, while other scenarios resulted in considerably lower stresses.

A fully dynamic fragility analysis was performed based on the favorable validation of the FE model. Two major scenarios were studied: M 6 earthquake with $R_{Rup}$ < 10 km on one of the potentially active faults of the Sinai Negev Shear Zone, and M 7 earthquake with $R_{Rup}$ = 45 km on the active Dead Sea Transform. For the M 6 earthquakes, the dynamic analysis yielded considerably higher basal stresses than the equivalent cylinder analysis. The dynamic stresses exceed the basal strength of the pillar, by a factor of two or higher.

Based on our findings, we postulate that the M 6 scenario on the SNSZ should lead to breakage of the Ramon pillar at its base due to tensile stresses exceeding its strength. Conservatively assuming that the first exceedance does not lead to pillar collapse, but does change the mechanical behavior from cantilever to PBR, it predicts a toppling failure during the next M 6 earthquake. With a fragility age of 11.4 ky, our analysis challenges the assumption that the SNSZ as a whole can produce an M 6 event.

## Appendices

**TableA1. Schmidt hammer elastic modulus survey data. Averaging mode ASTM.**

| Line stop | Line (m) | Line corrected (m) | measurement # | Time | Rebound value | E_avg (Gpa) | STDV (GPa) |
|---|---|---|---|---|---|---|---|
| 17 | 42.4 | 41.8 | 331 | 13:01 | 34.5 | 6.0 | 0.9 |
| 16 | 39 | 38.4 | 321 | 12:58 | 28.7 | 4.0 | 0.9 |
| 15 | 36 | 35.4 | 311 | 12:54 | 50.7 | 17.6 | 1.2 |
| 14 | 31.3 | 30.7 | 301 | 12:44 | 33.3 | 5.5 | 1.3 |
| 13 | 30 | 29.4 | 291 | 12:35 | 47.5 | 14.2 | 1.7 |
| 12 | 28.6 | 28 | 281 | 12:26 | 64.0 | 43.0 | 1.0 |
| 11 | 28.3 | 27.7 | 271 | 12:24 | 57.4 | 27.6 | 1.3 |
| 10 | 25.3 | 24.7 | 254 | 12:05 | 53.3 | 21.0 | 1.1 |
| 9 | 24.5 | 23.9 | 243 | 12:02 | 50.6 | 17.5 | 1.0 |
| 8 | 24 | 23.4 | 232 | 11:58 | 51.9 | 19.1 | 0.8 |
| 7 | 21 | 20.4 | 221 | 11:55 | 45.9 | 12.8 | 1.0 |
| 6 | 19 | 18.4 | 210 | 11:51 | 66.7 | 51.5 | 0.7 |
| 5 | 16 | 15.4 | 200 | 11:45 | 35.0 | 6.2 | 0.9 |
| 4 | 12 | 11.4 | 186 | 11:41 | 57.0 | 26.9 | 1.0 |
| 3 | 9 | 8.4 | 176 | 11:36 | 59.4 | 31.7 | 1.0 |
| 2 | 6.7 | 6.1 | 166 | 11:32 | 53.8 | 21.8 | 1.1 |
| 1 | 1.7 | 1.1 | 156 | 11:31 | 63.9 | 42.8 | 0.9 |

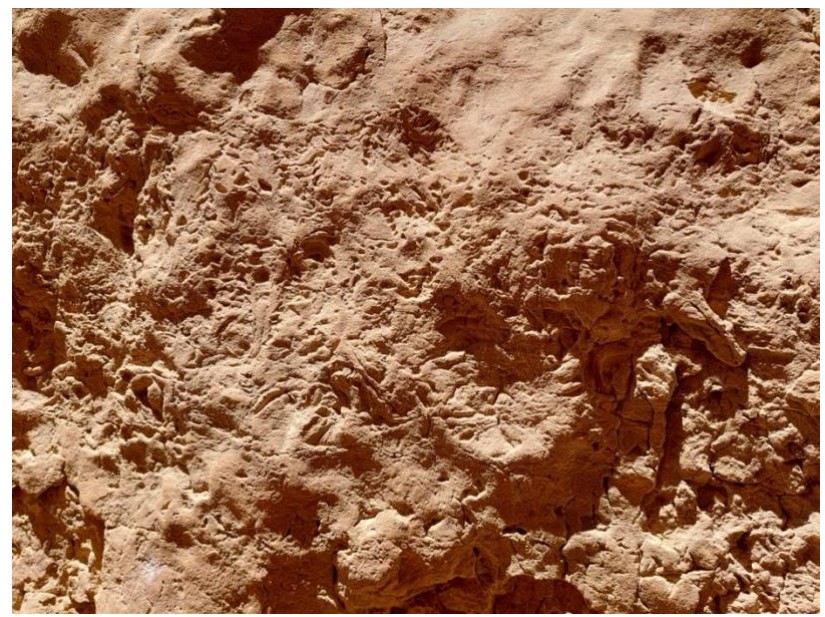

**Figure A1. The transition from massive limestone to a porous fossilized reef at a back-crack depth of 15 m**.

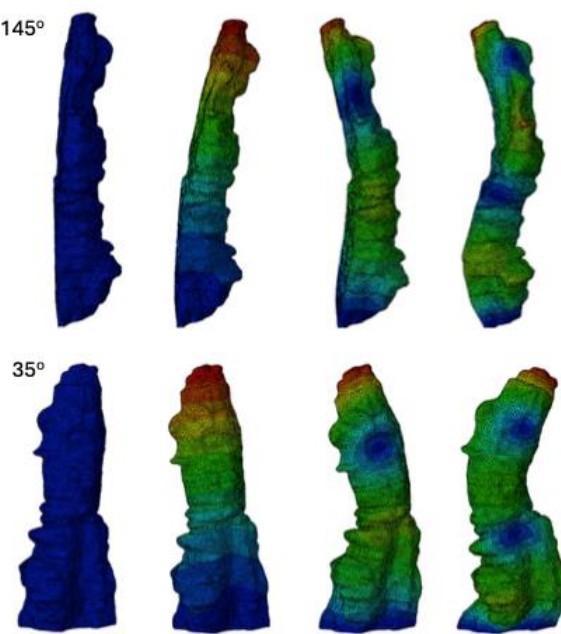

**Figure A2. Visualization of the Abaqus FE modal analysis of Scan_H model, first 3 modes. Top is145º direction and bottom 35º** 
**direction. Color scales for each direction and modes are for different values of modal displacement: blue (zero) to red (maximum). For the 145º direction, the maximum values are 1.166 ·10⁻³ m, 1.286 ·10⁻³ m, and 1.480 ·10⁻³ m (first, second, and third modes respectively). For the 35º direction, maximum values are 1.190 ·10⁻³ m, 1.154 ·10⁻³ m, 1.073 ·10⁻³ m (first, second, and third modes respectively).**

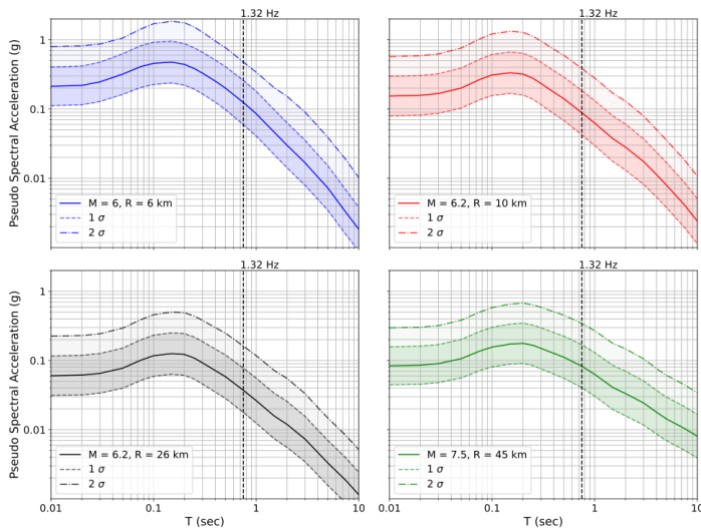

**Figure A3. Pseudo-spectral accelerations for the different fault–distance scenarios based on the ASK14 (Abrahamson et al., 2014) ground motion model. The continuous line is the median value, and the shaded region is one standard deviation. 1.32 Hz is the first mode of the Ramon pillar.**

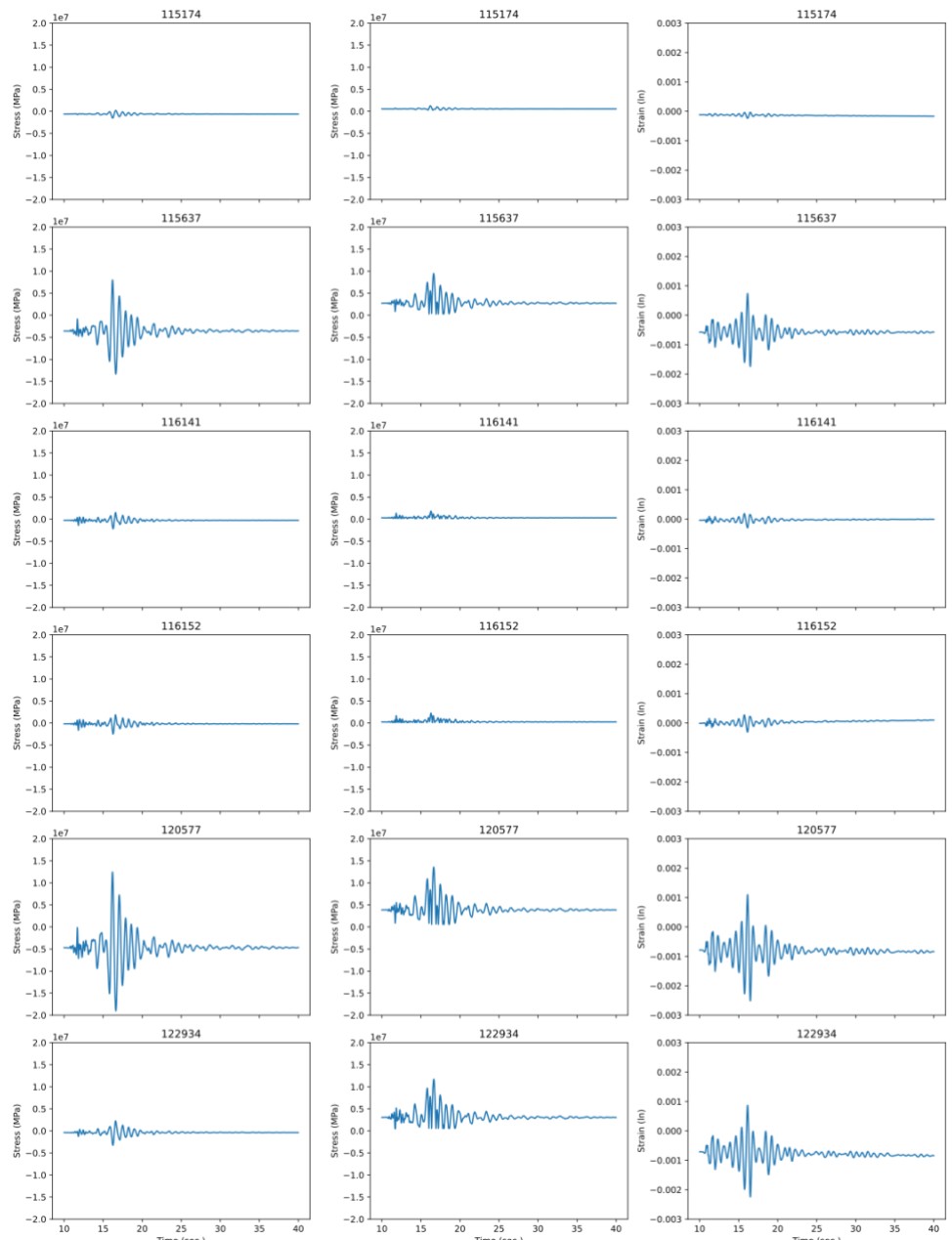

**Figure A4. Stress and strain outputs for selected elements at the pillar's base for selected elements. The left column represents vertical stress (S22), the central column represents Tresca stress, and the right column represents logarithmic strain. Values are plotted for the dynamic loading phase.**

**Data Availability and Resources**

Israel Bulletin Earthquake data is available at https://eq.gsi.gov.il/en/earthquake/accEarthquakes.php (last accessed 24.12.2024). Rock mass data will be made available by request from the corresponding author.

**Authorship contribution statement**

A. Jbara: Investigation, Formal analysis, Writing. M. Tsesarsky: Conceptualization, Funding acquisition, Supervision, Methodology, Investigation, Writing.

**Declaration of competing interest**

The authors declare that they have no known competing interests that could have influenced the work reported in this paper.

**Acknowledgments**

This research was partially sponsored by the Israel Science Foundation, grant 1163/22. We thank Boaz Langford for his assistance with rappelling and in-situ measurements.

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
