# Peer review of "Dynamic Fragility of a Slender Rock Pillar in a Sedimentary Rock Mass – from rock mechanics to seismic hazard"

_Natural Hazards and Earth System Sciences, 2024_

## Author Comment (AC1)

**Authors Reply "Dynamic Fragility of a Slender Rock Pillar in a Sedimentary Rock Mass"**

**nhess-2024-150**

We sincerely appreciate the time and effort the two anonymous reviewers have devoted to evaluating our manuscript. Their insightful comments and constructive feedback have been invaluable in enhancing the clarity and quality of our work. We are grateful for their thoughtful suggestions, which helped us refine our analysis and improve the paper. Below, we provide detailed responses to each point raised and outline the revisions to address their concerns. Our response is marked by blue colored text

**RC1**

This manuscript provided a comprehensive analysis of the dynamic fragility of a slender rock pillar (Ramon, Israel) based on accurate LiDAR scanning of its geometry, in-situ rock elastic modulus determination, and FEM modal and dynamic analysis. Nonetheless, the details of model construction are unclear, and there is a lack of corresponding theoretical equations and measured data, so I do have some comments which I believe will improve the impact of the work. Anyway, it is recommended for publication in NHESS after the following major revision to improve its quality before publication:

1. The abstract is not enough to summarize the research results of the manuscript, please revise the abstract again.

   **1R**: Thank you for pointing this out. We have thoroughly revised the abstract better to convey our study's key findings and significance.

2. In the Introduction, the significance and innovation of the research in this paper are not clearly explained, and the current research results of dynamic brittleness in the slender rock pillar are not explained, to reflect the research significance and innovation of the manuscript, please revise the introduction.

   **2R**: We have rewritten the relevant part of the Introduction for enhanced clarity, lines 85--90.

3. In section 3.1, there is a lack of measurement point layout method and measurement error for the elastic modulus of the Ramon Pillar, please supplement the measurement point layout scheme and on-site real picture.

**3R:** The initial plan for in-situ measurements of the rebound values (Schmidt Hammer) measurements was to achieve "evenly spaced" measurements. However, the reality was more complicated. The first two meters were on an overhanging rock ledge, and rappelling down required a few meters of rope slack so we could reach the wall. Moreover, rappeling along the length of the pillar detachment crack was complicated because, in several locations, "reaction" support by the rappeling partner was required. Hence, locations were not regularly spaced. We tried to capture lithological variations, s in the carbonate rock mass, structural and compositional, for example, from massive limestone to porous reefs. The outcome was the uneven spacing of measurements. Geological judgment, based on the measurements and visuals, was applied to the division. We add a short description on lines 141 - 147 and a photo in the appendix (Figure A1) as an example of variations.

4. In section 3.2, only the test data of elastic modulus are given during the construction of the numerical model, but other parameters of lithology are not given, such as Poisson's ratio, please add.

**4R:** Added as suggested. The "FEM Analysis" section was rewritten.

5. In section 3.2, the details of model construction are not clear, such as the boundary condition setting method, the load application method, and the related mathematics and mathematical models of the model.

**5R:** We added more details on boundary conditions and load application methods.

6. The manuscript only uses LiDAR to scan the outer surface of the Ramon Pillar, but the trend of the internal fractures and bedding planes of the rock has a great influence on the tensile strength of the rock. Does the numerical model construction process consider the influence of the trend of the internal fractures and bedding planes of the Ramon Pillar, and how is it realized?

**6R**:  In this study, the influence of internal fractures was not explicitly modeled. However, we acknowledge their significant contribution, along with bedding planes, to the overall deformability and strength of the rock mass (e.g., Shang et al., 2018; Frayssines and Hantz, 2009; Elmo et al., 2018). To account for these effects, the rock mass properties were adjusted using the Geological Strength Index (GSI) and the damage parameter $D$ (Hoek and Diederichs, 2006). The rock mass modulus ($E_{RM}$) was validated by comparing the calculated and measured natural vibration modes of the pillar. The validation was successful, with the primary vibration modes accurately represented. Furthermore, the estimated rock mass damping factor, derived from measured vibrations, indicated

relatively high damping (5% to 6%), likely attributable to multiple discontinuities within the rock mass. We added clarification on lines 134-137.

7. In the 4. Result analysis part, the display and analysis of the simulation results of the model are not in-depth, and there is a lack of stress, strain and plastic zone data, pictures and corresponding correlation analysis after the model is calculated. Dig into the model calculations and analyze them in depth to argue the arguments of the manuscript.

**7R:** The analysis was intentionally kept simple, resulting in a succinct presentation. The results section was expanded, focusing on stresses, allowing for comparison with the material properties. We added an example figure in the appendix (Figure A4) to illustrate stresses and strains for selected elements at the pillar's base.

8. In the discussion section, there is also a lack of relevant model calculation data and cloud maps, which need to be supplemented.

**8R:** We have expanded the section on the simplified modal analysis (4.2) to provide a more detailed explanation of the use of Ground Motion Models (GMM) (lines 210– 220) and the dynamic analysis (lines 260– 270). However, we are unclear about the "cloud maps" referenced by the reviewer. Could the reviewer kindly provide clarification or further details on this point to ensure we address this concern appropriately?

9. The references are too outdated, with papers from 10 years ago accounting for as high as 46.34% of the total. Please update to the latest references.

**9R:** We have included more up-to-date references, particularly those focused on the applications of precariously balanced rocks (PBRs), as they are more prevalent in the literature. The relatively high proportion of older references pertains to regional seismic hazard assessments, NGA-2 West Ground Motion Models (primarily from 2014), and fundamental rock mechanics, all of which are directly relevant to the fragility analysis of the Ramon Pillar.

10. The manuscript could benefit from a refinement in writing style. For instance, it is advisable to minimize the use of first-person pronouns like "we" and active expressions such as "We studied" or "We defined." Employing the passive voice consistently across the paper would be more appropriate.

**10R**: We accept the suggestion and use passive voice across the manuscript.

In conclusion, I trust my feedback will be beneficial, and I anticipate the opportunity to assess the refined and enhanced manuscript.

**It deed. Thank you for the time and effort.**

**RC2**

The manuscript of Jbara and Tsesarsky presents the fragility analysis of a fragile geologic feature (FGF), specifically a 42 m rock pillar that is located in the Negev Desert, for the purpose for making inferences about the maximum magnitude of ancient earthquakes on the nearby Sinai Negev Shear Zone. The authors combined high-resolution LiDAR with measurements of rock elastic modulus along the height of the pillar to produce a finite element model of the pillar. Importantly, they showed that both a simplified geometry or a simplified assumption of rock elastic modulus yielded inaccurate results. A dynamic fragility analysis of the rock pillar was then performed using ground-motion time histories of magnitudes and distance of relevant to their rock pillar. Their primary result is the inconsistency of a M6 earthquake on the Sinai Negev Shear Zone and the observed survival of the feature over its 11.4 ky fragility age. This publication will be a valuable addition to the actively growing research into using FGFs to test seismic hazard models and their components, particularly as this manuscript focuses on expanding the use of the FGF class of rock pillars.

**Reviewer Comments:**

Regarding the title, seismic hazard is the rate or probability of exceeding a level of ground-motion intensity at a site. While the author's do make comparisons to a ground-motion model and comment on a relevant seismic source to the site, the claim of "to seismic hazard" is not accurate. I would suggest rewording, as this work tests the magnitude of potential earthquakes on local active faults.

**Reply:** We thank the reviewer for this remark. Our use of "seismic hazard" terminology is indeed not accurate. Accordingly, we change the title to "Dynamic Fragility of a Slender Rock Pillar in a Carbonate Rock Mass"

My major comments relate to the need for several key concepts to be corrected or reworded to avoid reader confusion.

First, FGFs do not "validate seismic hazard analysis", it is not the framework that is validated but a given PSHA model/results/estimates that are validated.

**Reply:** Thank you for correcting our inconsistent and inaccurate use of terminology. We have changed the abstract accordingly. Lines 35 - 36.

Second, rewording is required in a couple of places to make it clearer that it's not the case the PBRs are geomorphically the most common class of FGFs to form, but that precariously balanced rocks (PBRs) are the class of FGFs that have been investigated the most for the purpose of putting constraints on seismic hazard estimates.

**Reply**: We corrected this ambiguity as suggested.

Third, it is correct to say that the PBR fragility methods cannot be directly applied to other FGF classes that have different failure modes, i.e., beam breaking. However, the PBR fragility methods can be applied to PBRs of any lithology (metamorphic: e.g., Stirling et al., 2012, igneous: e.g., Rood et al., 2024, and sedimentary: e.g., Rood et al., 2020) and for PBRs formed in any geographical location. It is the geometry and mechanics of the rock pillars that requires breaking fragilities (rather than rocking and toppling fragilities) to be determined. Importantly, it is not due to them being composed of sedimentary rocks (the rock pillars could be composed of any rock type, but their mode of failure would be the same). Please correct this throughout.

**Reply**: We agree with this remark, and we have rewritten the first paragraph of section 1.2. Lines 62 – 66.

Based on Figure 3, 17 rebound hammer readings were taken over the 42 m height and the readings were not regularly spaced. Please include a description of the criteria used for the selection of measurement locations, as well as whether an increased density of samples (for example every meter, or sub-meter spacing) would be expected to improve or change the results in a significant way. Also, can you please provide a couple of examples of the "geological judgment" that were used to divide one region from another, as the differences between the divisions are not super obvious based on the data in Figure 3 alone.

**Reply**: Please see our reply to point 3 of reviewer #1.

For the "Simplified Fragility Analysis" what is the justification for using the Abrahamson, Silva, and Kamai (2014) ground-motion model, rather than: Chiou and Youngs (2014) global ground-motion model or the Akkar, Sandıkkaya, and Bommer (2014) Europe/Middle East ground-motion model (these would seem to be the more obvious choices). Please clarify the choice.

**Reply**: A comparison of the NGA-West 2-based GMMs (Gregor et al., 2014) shows great similarity for M6 and M7 earthquakes. ASK14 and CY14 practically overlap (Fig. 1 there). Only for M8 do the GMMs diverge, but this is mainly due to a limited number of events. As per ASB14, it is based on 221 events and 1041 records, compared with 300 events and 12,000 records for ASK14 or CY14. Comparison of ASB14 with the 2008 version of CY (or equivalent),

shows small differences (Fig. 11 in ASB14). We chose ASK14 as a representative GMM for analysis. We clarify this choice in lines 210 - 220.

Additionally, if there is the truncation of ground-motion models in probabilistic seismic hazard analysis, it is likely at 3 or 5 sigma. Showing 1, 2, and 3 sigma with different shading/line pattern would be helpful on the Figure A2 plots to show the uncertainty included in these models. Or this point should at least be included in the text.

**Reply**: Thank you for this valuable remark. Truncation (sigma choice) is an important issue in PSHA and DSHA analyses. In DSHA, it is typically assumed that $1\sigma$ (84th percentile) represents the maximum worst-case scenario' ground motion assumed to be the boundary between physically possible and unphysical ground motions (Strasser et al., 2008). Whereas such a limit is not imposed in PSHA the truncation depends on return periods. Assuming an annual frequency of exceedance of $10^{-3}$ (millennial return), the difference between $1\sigma$ and $2\sigma$ to $3\sigma$ is relatively small. We add a clarification in the text (lines 218 - 220). In the Discussion section, we further discuss this issue as $2\sigma$ truncation elevates the stress for the M6.2R10 (Nafha-Saad fault) above the basal tensile strength, lines 313 - 315. We also add the $2\sigma$ curves in Figure A4.

Finally, it also needs mentioning somewhere this is an ergodic ground-motion model, and FGFs should be compared to non-ergodic ground-motion estimates to account for the source, path, and site effects specific to the FGF.

Reply: We address this remark in the Discussion section, lines 315 - 317.

For the time histories used for the finite element analysis, it is important to point out that earthquakes of those magnitudes and distances have not been recorded for your specific faults and sites of interest, and that even in the global PEER database there are limited recordings of the relevant large magnitudes at short distances. Therefore, it is necessary to make the assumption that the recordings that you selected for your analysis are appropriate to be applied for your source, path, and site combination.

Reply: We address this issue in both the Results and Discussion sections.

Also, related to the finite element analysis, was the cliff face that can be seen in Figure 2 along one side of the rock pillar included in the modelling? Please clarify this in the manuscript, as well as explain what the effect of this cliff face is (if any) is on the fragility of the pillar. For example, I wonder whether the bending of the pillar as a cantilever beam before breaking would result in there being a collision with the cliff, dampening the response?

Reply: The cliff was not included in the finite element (FE) analysis. Clarification regarding this omission has been added to the FEM Analysis section (line 158). According to the

dynamic fragility modeling, the maximum displacement of the pillar top during bending, as shown in Table 4, was 9 cm (for the ChiChi earthquake loading). Considering the meters-scale distance between the cliff and the pillar, collision and dampening are not likely.

Regarding the conclusions about a M6 earthquake event on the Ramon and Nafha-Sa'ad faults not occurring over the past ~11,000 years, what is the rate of occurrence of a M6 earthquake from the magnitude-frequency distribution of these faults? Either from literature or hazard model seismic sources? Specifically, of interest is whether the magnitude-frequency distribution estimates a M6 event more or less frequently than ~11,000 years, as that would make the model inconsistent or consistent with your FGF analysis.

Reply: For the faults of the SNSZ, magnitude-frequency distribution is unknown. Although low-magnitude seismicity was recorded, it is sparse in space and time. Grunthal et al. (2009) assign a maximum value of 6.2 for the East-Sinai-Negev region in general. The same report provides frequency-magnitude relation for the entire Dead Sea seismic zone with a common -b for small seismic sources within the larger zone of 0.938± 0.022. for M an 6.2 earthquake, the annual rate is in the order of $10^{-2}$. We added the missing information in the Seismicity section and rewrote the entire section. Thank you for pointing out the missing data.

Also, given that the magnitude of a rupture and the ground-motion at a distant site are linked but separate, please add some discussion around the fact that the magnitude may be consistent with the survival of the rock pillar. For example, it is possible that the ground-motions from a M6 SNSZ rupture are lower at the rock pillar site than those of the small number of recordings in the PEER database or of the ergodic ground-motion model. Non-ergodic ground-motion models or physics-based simulations would help rule out this possibility.

Reply: We have addressed the ergodic assumption (following a comment above) in the Discussion section in the context of the GMMs. Naturally, the same logic applies to magnitude-ground motion-distance relations for a selected number of recordings from a global (ergodic) catalog. We are planning to perform physics-based wave propagation simulations in future research.

Finally, a detailed review of the manuscript focusing of consistent word choices and paragraph structure (there are lots of 2 sentence paragraphs!) would greatly benefit the readability. I believe that the clarifications and additional information requested in my review are minor revisions, and I would be happy to review the manuscript again after the necessary revisions have been made and prior to publication.

Reply: Thank you for the comment. We implemented the suggestion.

Figure Comments:

Figure 1: Need to provide a link or reference for the access to the Israel Seismic Catalog plotted. Also, scaling the size and/or colour of the catalog circle symbols with the magnitude of the earthquake event would be helpful for understanding the historical seismicity of the region.

Reply: A reference to the Israel Seismic Network (ISN) catalog was added to the Data and Resources section. As per magnitude, the presented events are for a 10-year period (2013 – 2023). The main purpose was to "highlight" the more active faults of the Dead Sea region. The ISN catalog for this period contains 2823 events with M > 2, of which only 30 (about 1%) are M > 4, and only two (not in the plotted region) are M 5. Differentiating events by color or size was avoided, as most events would share the same color, and such differentiation would introduce "visual noise" to the figure.

Figure 2: The look direction of the photos would be helpful in understanding the failure direction of the features relative to the orientation of the local faults.

Reply: Thank you. Added at the caption.

Figure A1 needs a colour bar scale.

Reply: each mode has a different scale for the color bar. We add an explanation and min/max values in the caption.

References

Gregor, N., Abrahamson, N. A., Atkinson, G. M., Boore, D. M., Bozorgnia, Y., Campbell, K. W., Chiou, B. S.-J., Idriss, I. M., Kamai, R., Seyhan, E., Silva, W., Stewart, J. P., and Youngs, R.: Comparison of NGA-West2 GMPEs, Earthquake Spectra, 30, 1179-1197, 10.1193/070113eqs186m, 2014.

Strasser, F. O., Bommer, J. J., and Abrahamson, N. A.: Truncation of the distribution of ground-motion residuals, Journal of Seismology, 12, 79-105, 10.1007/s10950-007-9073-z, 2008.